# Volumetric Plane-based Rendering: A Novel Approach for Internal View Synthesis in 3D Gaussian Splatting

Sanghyuk Roy Choi[1]    Minhyeok Lee[1,2,†]

[1]Department of Intelligent Semiconductor Engineering, [2]School of Electrical and Electronics Engineering

Chung-Ang University, Seoul, Korea

{choiroy, mlee}@cau.ac.kr

## Abstract

*3D Gaussian Splatting (3DGS) represents scenes as sets of anisotropic 3D Gaussians optimized from multi-view exterior images. Training 3DGS is largely surface-driven and provides limited gradient for the volume interior. We investigate whether 3D Gaussians can capture internal volumetric structure when training is conditioned on cross-sectional slices. We propose GS-IR, which replaces exterior-view supervision with plane-conditioned internal rendering. GS-IR places virtual cameras along a semicircular trajectory and defines a thick cross-sectional plane passing through the volume center. A binary mask selects only the Gaussians whose centers fall within the plane; the selected subset is rasterized with the standard alpha-blending pipeline and optimized against the corresponding ground-truth slice obtained by trilinear interpolation of the input volume. Cycling through viewpoints provides supervision to Gaussians throughout the volume during training. This simple gradient-routing mechanism enables stable optimization of interior Gaussians and complements standard densification and pruning across existing 3DGS variants. We evaluate GS-IR on the KiTS23 (CT) and IXI (MRI) datasets. GS-IR can be applied to 3DGS, 2DGS, and Mip-Splatting without modifying their Gaussian rasterizers. It improves PSNR by up to +21.52 dB and SSIM by up to +0.54 over the respective baselines.*

## 1. Introduction

Neural Radiance Fields (NeRF) [15] have emerged as a powerful framework for high-quality novel-view synthesis. 3D Gaussian Splatting (3DGS) [12] advances this line of work by representing scenes as explicit anisotropic Gaussians and rasterizing them with an efficient tile-based pipeline, achieving superior speed and quality over prior volumetric methods.

This efficiency and fidelity have driven a rapid expansion of variants. Mip-Splatting [24] addresses scale-dependent aliasing. 2D Gaussian Splatting (2DGS) [10] improves geometric accuracy through oriented 2D Gaussians. Gaussian-based SLAM systems such as SplaTAM [11] extend 3DGS to online dense mapping and localization.

Neural rendering has also been applied to projection-based medical imaging. MedNeRF [4] synthesizes Digitally Reconstructed Radiographs (DRRs) from sparse X-ray inputs using NeRF-based representations, while X-Gaussian [3] and $R^2$-Gaussian [25] adapt 3DGS to X-ray image and sparse-view CT reconstruction, respectively. These methods operate under projection-based observation models.

Despite these advances, existing NeRF and 3DGS-based methods are primarily designed for camera-visible appearance and surface geometry rather than direct supervision of internal volumetric structure. In standard 3DGS, adaptive densification concentrates Gaussians around regions that contribute to observed views, which makes internal slice reconstruction from volumetric supervision challenging.

We investigate whether Gaussians can be adapted to represent the internal volumetric structure beyond surface geometry. Evaluating this question requires volumetric data with known internal anatomy. CT and MRI satisfy this requirement because they provide dense 3D volumes from which cross-sectional slices can be extracted as ground-truth supervision.

We present Gaussian Splatting for Internal Representation (GS-IR), a framework based on volumetric plane-based rendering. GS-IR initializes Gaussians densely throughout the volume and places $N$ virtual cameras along a semi-circular trajectory, each defining a cross-sectional plane of predefined thickness through the volume center. At each training step, a binary mask extracts the Gaussians within the corresponding plane; this subset is rasterized via alpha-blending and optimized against the corresponding ground-truth slice obtained by trilinear interpolation of the original volume. Gradients are backpropagated to the selected Gaussians leaving the remaining Gaussians frozen.

---

[†]Corresponding author.

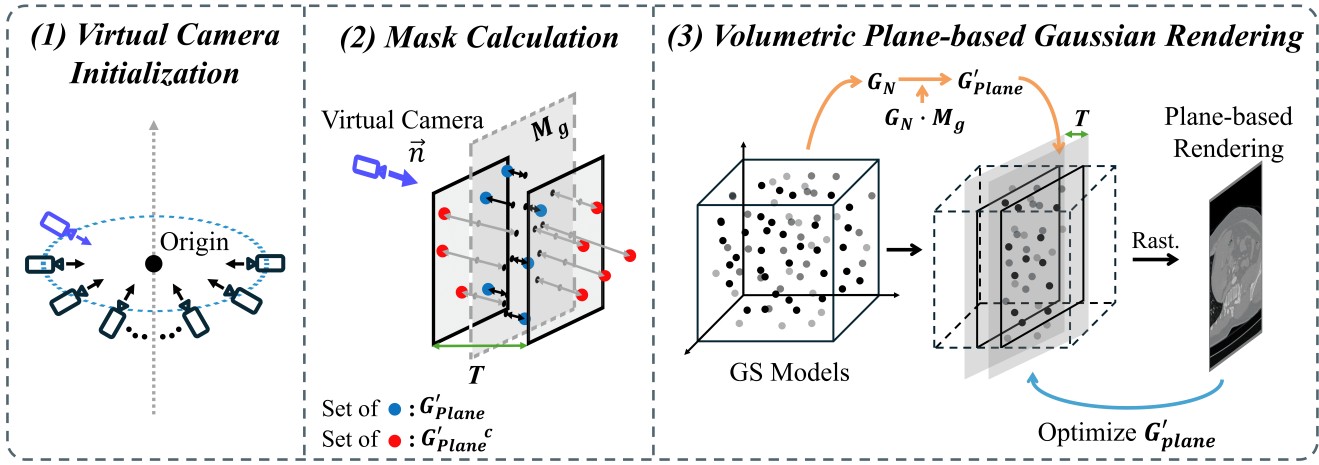

Figure 1. Overview of our proposed GS-IR pipeline. Our framework defines virtual camera poses arranged uniformly along a semicircular trajectory, with all cameras oriented towards the origin. A sample camera highlighted in blue determines the normal vector $n$ of a volumetric plane. The distances from this plane to all Gaussians are then calculated; Gaussians within a distance threshold of $T/2$ indicated by the green arrow are extracted while the remaining Gaussians are masked. In the masking illustration, the red dots denote the selected Gaussian subset $G'_{\text{plane}}$, whereas the blue dots represent the excluded set of Gaussians $G'_{\text{plane}}{}^c$. The subset $G'_{\text{plane}}$ is derived by applying $M_g$ to the full Gaussian set $G_N$ and rasterized to produce a rendered image. This rendered image is compared against the ground truth and the loss is calculated. The Gaussians in $G'_{\text{plane}}$ are optimized using this loss.

Because the planes are uniformly distributed over the $180°/N$, Gaussians across the volume receive supervision from one or more planes during training. GS-IR therefore enables Gaussians to learn internal anatomy without modifying the underlying rasterizer.

## 2. Related Work

### 2.1. Neural Radiance Fields for Medical Imaging

The success of NeRF [15] in novel-view synthesis has motivated its extension to medical imaging. MedNeRF [4] builds on Generative Radiance Fields (GRAF) [21] to synthesize digitally reconstructed radiographs (DRRs) from extremely sparse X-ray inputs. Its generator samples points along camera rays to predict density and attenuation, while a discriminator compares the rendered patches against real DRR patches in an adversarial training framework.

### 2.2. Variants of 3D Gaussian Splatting

A broad family of methods has extended 3DGS along multiple axes. For rendering fidelity, Mip-Splatting [24] introduces 3D smoothing filters that apply a low-pass filter to the Gaussian primitives prior to projection, suppressing scale-dependent aliasing across varying camera distances. Analytic-Splatting [14] employs per-pixel analytic integration over the Gaussian footprint to reduce shimmering artifacts under camera motion. For geometric accuracy, 2DGS [9] replaces volumetric Gaussian kernels with oriented 2D Gaussians that collapse one scale axis

to zero, enforcing a surface-aligned representation and enabling cleaner mesh extraction.

### 2.3. Gaussian Splatting for Medical Imaging

Recent work has extended 3DGS [12] to medical transmission imaging by redesigning the rendering pipeline to account for X-ray physics. X-Gaussian [3] replaces the view-dependent spherical harmonics of standard 3DGS with a Radiation Intensity Response Function that models the isotropic nature of X-ray imaging, and develops a Differentiable Radiative Rasterization kernel for efficient novel-view X-ray synthesis. $R^2$-Gaussian [25] extends this approach by introducing a radiative Gaussian kernel parameterized by central density, position, and covariance, while rectifying the rasterizer to mitigate the integration bias of standard 3DGS in tomographic reconstruction. It further incorporates a differentiable voxelizer that converts optimized Gaussians into a density volume, enabling direct 3D volume retrieval from sparse-view X-ray projections. 3DGR-CT [13] addresses sparse-view CT by initializing Gaussians from FBP-reconstructed images to avoid placing primitives in void regions, and integrates them with a differentiable CT projector through NeRF-style volumetric rendering. Vol3DGS [22] proposes volumetrically consistent rasterization by analytically computing transmittance along camera rays rather than relying on the splatting approximation, improving physical accuracy for both view synthesis and tomographic reconstruction. For MRI, 3DGSMR [18] employs 3D Gaussian distributions as an explicit represen-

tation for MR volumes, reconstructing isotropic resolution 3D MRI from undersampled k-space data under a self-supervised framework.

## 3. Method

We present GS-IR to mitigate the surface bias of standard Gaussian Splatting under slice-based supervision and to perform internal volumetric reconstruction within the Gaussian Splatting framework. We evaluate GS-IR on CT and MRI volumes, where it learns coherent internal representations from cross-sectional supervision.

### 3.1. Virtual Camera Initialization

**Challenge of Volumetric Data.** Volumetric medical images such as CT and MRI pose a significant challenge for the application of the conventional Structure-from-Motion (SfM) algorithm [17]. While SfM pipelines typically derive camera parameters and generate a point cloud from a set of images that capture a scene from various viewpoints, medical data are fundamentally constrained to orthogonal views along the x, y, and z axes [1, 8, 20]. Consequently, the direct application of SfM to such data yields suboptimal results due to the lack of sufficient viewpoint variation.

**Virtual Camera Initialization.** To overcome this limitation, we introduce a virtual camera parameter initialization method that emulates the output of an SfM algorithm. We arrange 100 virtual cameras at equidistant intervals along a semicircular trajectory, all oriented toward the origin. As shown on the left of Fig. 1, all cameras within this configuration are precisely oriented to face the origin. We sample virtual cameras along a semicircular trajectory to expose the model to a diverse set of slice orientations. Each camera defines a unique slicing plane, allowing GS-IR to route supervision to the subset of Gaussians that represents the slice.

### 3.2. Volumetric Plane Definition

Each virtual camera defines a volumetric slicing plane utilized to select the relevant Gaussians. The volumetric plane is defined by its position, normal vector, and thickness. In our implementation, each volumetric plane passes through the origin of the point cloud. The normal vector of the volumetric plane is aligned with the corresponding camera viewpoint.

**Plane Thickness Definition.** A critical component of our method is the incorporation of a uniform thickness transforming the 2D plane into a volumetric plane. An appropriate value of thickness is critical: an insufficient thickness results in an inadequate number of Gaussians being included, leading to artifacts, whereas excessive thickness causes overlap with volumetric planes from adjacent viewpoints, degrading prediction accuracy.

Therefore, we determine the thickness by calculating a value that minimizes overlap between adjacent planes while maintaining sufficient Gaussian coverage. The thickness value $T$ is defined as:

$$T = (L/2) \cdot \sin(180°/N), \tag{1}$$

where $T$ represents the thickness of the plane; $L$ denotes the side length of the cube-shaped 3D point cloud; $N$ represents the total number of virtual cameras. The thickness $T$ is computed from the angular separation $180°/N$ between the adjacent planes and the maximum radius $L/2$ of the point cloud. This formulation ensures that the volumetric plane thickness scales appropriately with the scene size and camera density.

### 3.3. Volumetric Plane-based Gaussian Rendering

**Gaussian Mask Calculation.** To extract the Gaussians that lie within the volumetric plane, a binary mask vector $M_g$ is computed for each Gaussian $g_i$. A masking operation is performed to selectively extract the Gaussians that intersect with the given volumetric plane as defined in the following equation:

$$M_g(i) = \begin{cases} 1, & \text{if } |(\boldsymbol{\mu}_i - \mathbf{p}_0) \cdot \mathbf{n}| \leq \dfrac{T}{2}, \\ 0, & \text{otherwise,} \end{cases} \tag{2}$$

where, $M_g(i)$ represents the resulting binary mask value for the $i^{th}$ Gaussian whose center position is denoted by $\mu_i$; $p_0$ represents the origin on the central plane of the 3D point cloud; $n$ denotes the unit normal vector of the slicing plane and is aligned with the corresponding camera viewing direction; $T$ is the hyperparameter thickness of the volumetric plane.

**Gaussian Extraction.** The masking operation using mask vector $M_g$ calculated by Eq. 2 can be represented by the following equation:

$$G'_{\text{plane}} = \{g_i \in G_N \mid M_g(i) = 1\}, \tag{3}$$

where, $G'_{\text{plane}}$ represents a set of extracted Gaussians; $G_N$ represents a set of total Gaussians; $N$ represents the total number of Gaussians, and $M_g \in \mathbb{R}^{1 \times N}$ represents a masking vector for extracting Gaussians within the volumetric plane. We compute and apply $M_g$ to $G_N$ to create $G'_{\text{plane}}$ for each camera viewpoint.

**Modified Rendering Equation with Mask.** We adapt the standard 3DGS alpha-blending equation for our model by rendering exclusively from the extracted Gaussian subset $G'_{\text{plane}}$. Our modified rendering equation $C_{\text{Plane}}$ can be represented by the following equation:

$$C_{\text{Plane}} = \sum_{i \in G'_{\text{plane}}} c_i \alpha_i \prod_{j=1}^{i-1} (1 - \alpha_j), \tag{4}$$

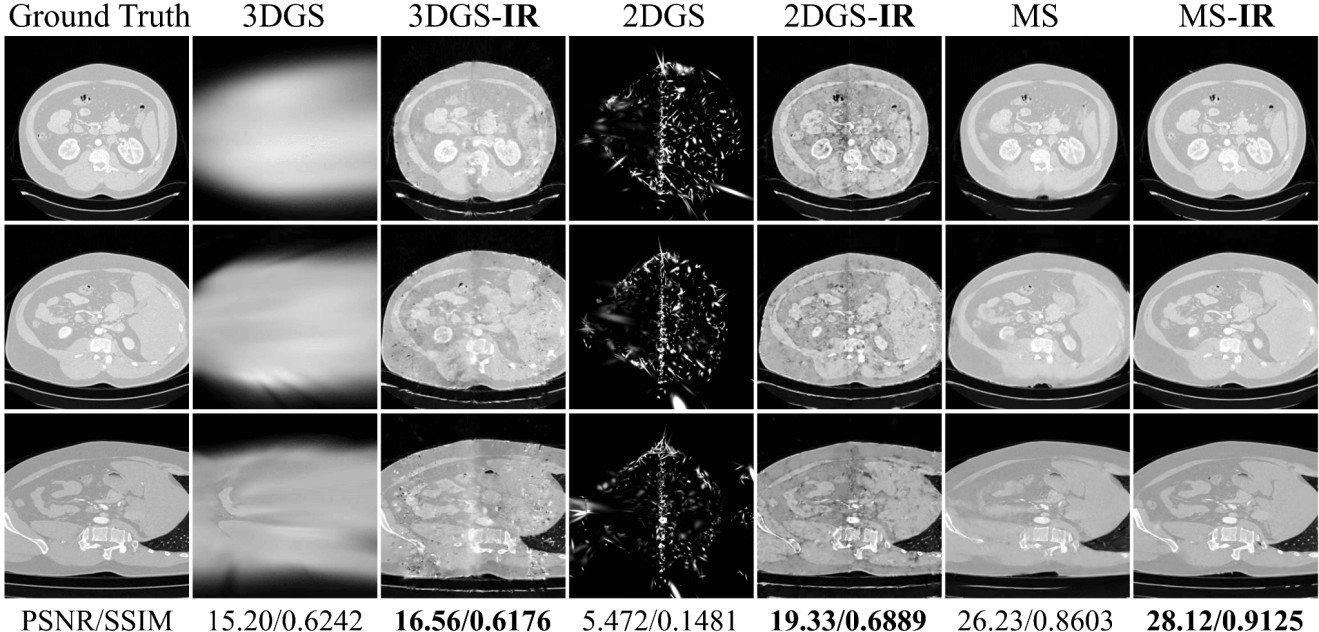

| | Ground Truth | 3DGS | 3DGS-**IR** | 2DGS | 2DGS-**IR** | MS | MS-**IR** |
|---|---|---|---|---|---|---|---|
| PSNR/SSIM | | 15.20/0.6242 | **16.56/0.6176** | 5.472/0.1481 | **19.33/0.6889** | 26.23/0.8603 | **28.12/0.9125** |

Figure 2. Qualitative and quantitative results on KiTS23 (CT). Columns (left to right) show Ground Truth, 3DGS, 3DGS-IR, 2DGS, 2DGS-IR, Mip-Splatting (MS), and MS-IR. The suffix -IR denotes the application of our proposed method. PSNR/SSIM are reported below each method column.

where $i, j$ denote indices in depth-sorted order within $G'_{\text{plane}}$. Eq. 4 adapts the standard 3DGS alpha-blending process by operating exclusively on the Gaussian subset $G'_{\text{plane}}$. This formulation ensures that only the colors $c_i$ and the opacities $\alpha_i$ of the Gaussians in $G'_{\text{plane}}$ contribute to the final color of the pixel $C_{\text{plane}}$. Gaussians outside this plane are completely excluded from the computation, receiving no gradient updates for this viewpoint.

The $G'_{\text{plane}}$ is rendered into a 2D cross-sectional image using the differentiable tile-based rasterizer adopted from the 3DGS framework [23]. The Gaussians in $G'_{\text{plane}}$ are projected onto the 2D image plane. These projected 2D splats are sorted based on the depth of the view-space. The tile-based alpha-blending is performed in parallel to synthesize the final 2D cross-sectional image, $I_{\text{p}}$.

### 3.4. Training Method

**Selective Optimization via Masking.** The rendered image $I_{\text{p}}$ is then compared with the corresponding ground-truth $I_{\text{gt}}$. Inherited from 3DGS, we use a weighted combination of $L_1$ loss and $D\text{-}SSIM$ loss weighted by hyperparameter $\lambda$. The loss function is represented in the following equation:

$$\mathcal{L}_{\text{mask}} = (1 - \lambda)\,\mathcal{L}_1(I_{\text{gt}}, I_{\text{p}}) + \lambda\,\mathcal{L}_{D\text{-}SSIM}(I_{\text{gt}}, I_{\text{p}}). \quad (5)$$

The Gaussian parameters in $G'_{\text{plane}}$ such as the 3D position, covariance, opacity, and SH coefficients are optimized

with the gradients computed from Eq. 5. By formulating the loss function as $\mathcal{L}_{\text{mask}}$, we ensure that the gradients are computed and backpropagated to the parameters $\theta_{\text{plane}} \subset \theta$ corresponding to the Gaussians $g_i \in G'_{\text{plane}}$. All other Gaussians $g_j \notin G'_{\text{plane}}$, where $M_{g_j} = 0$, are excluded from the rasterization and receive no gradient updates for the camera viewpoint. By iterating the selective optimization process across viewpoints, our model learns a coherent internal representation of the internal anatomy.

**Adaptive Density Control.** We employ the adaptive density control mechanism of 3DGS. The positional gradients used for densification are evaluated for the $G'_{\text{plane}}$. Similarly, pruning of Gaussians with low opacity is also applied within $G'_{\text{plane}}$.

### 3.5. Learning the Internal 3D Representation

Our training procedure iterates over a diverse set of virtual viewpoints rather than relying on sparse viewpoints. We employ an iterative optimization that cycles through the entire set of predefined virtual camera poses that span a semicircular trajectory $180°$. This approach encourages the model to learn a coherent 3D representation across the predefined volumetric planes from all angles.

By iterating the selective optimization process over the full distribution of virtual camera poses, our model progressively refines all spatial locations. This slice-by-slice mechanism ensures that a coherent 3D representation of the internal anatomy is learned, as each Gaussian is updated by

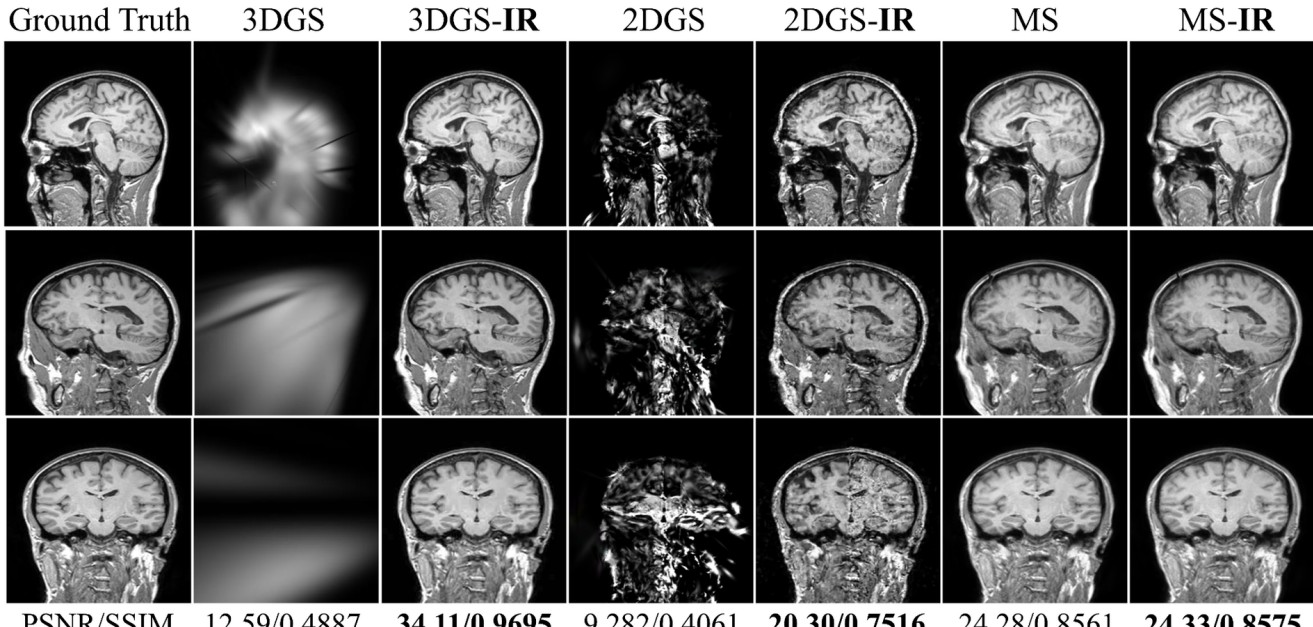

| | Ground Truth | 3DGS | 3DGS-**IR** | 2DGS | 2DGS-**IR** | MS | MS-**IR** |
|---|---|---|---|---|---|---|---|
| PSNR/SSIM | | 12.59/0.4887 | **34.11/0.9695** | 9.282/0.4061 | **20.30/0.7516** | 24.28/0.8561 | **24.33/0.8575** |

Figure 3. Qualitative and quantitative results on IXI (MRI) between Gaussian splatting baselines and their GS-IR variants. From left to right: ground truth, 3DGS, 3DGS-IR, 2DGS, 2DGS-IR, Mip-Splatting (MS), and MS-IR. The suffix -IR denotes the application of our proposed method. PSNR and SSIM are reported at the bottom of each method column.

multiple viewpoints that intersect its location.

## 4. Experiments

### 4.1. Experimental Setup

**Datasets.** The Kidney and Kidney Tumor Segmentation 2023 (KiTS23) dataset [8, 16] consists of abdominal CT volumes with an in-plane resolution of $512{\times}512$ and varying slice counts. We resize each volume to $512{\times}512{\times}512$ following prior work [5, 7]. The Information eXtraction from Images (IXI) dataset [2] consists of T1-weighted brain MRI volumes with a native resolution of $256{\times}256$ and varying slice counts. We resize each volume to $256{\times}256{\times}256$ using the same procedure [6, 19].

**Training Data Generation.** For both datasets, we generate 100 virtual camera viewpoints along the $180°$ semicircular trajectory described in Figure 1. The ground-truth image for each viewpoint is synthesized by applying trilinear interpolation to the volumetric data along the corresponding oblique plane, producing training images of $512{\times}512$ for KiTS23 and $256{\times}256$ for IXI.

**Implementation Details.** All models are trained and evaluated on a single NVIDIA RTX A6000 GPU with 48 GB of VRAM. For our GS-IR variants, training on each dataset takes approximately 1.5 hours. This runtime reflects the need to learn and refine Gaussians throughout the 3D interior volume under plane-conditioned supervision, rather than concentrating updates only on surface-visible regions.

We optimize all trainable parameters using Adam. Following the original 3DGS implementation, we set the learning rates to $1.6 \times 10^{-4}$ for Gaussian coordinates ($\mu$), $5 \times 10^{-2}$ for opacity ($\alpha$), $5 \times 10^{-3}$ for scaling ($s$), and $1 \times 10^{-3}$ for rotation ($q$).

**Evaluation Settings.** All backbone comparisons are conducted under an identical training setup, with the same training slices, virtual camera configuration, optimizer settings, and volumetric initialization. The only algorithmic variation is whether plane-conditioned Gaussian selection is used during rendering and optimization. Reconstruction quality is evaluated using four standard metrics: PSNR, SSIM, LPIPS, and MSE.

### 4.2. Internal View Synthesis on KiTS23 (CT)

**Qualitative Results.** Fig. 2 presents the qualitative comparison on the KiTS23. The standard 3DGS model produces a diffuse output that fails to resolve internal anatomy. The 2DGS baseline generates sparse and incoherent images dominated by artifacts and noise. In contrast, all three models augmented with GS-IR produce substantially sharper cross-sectional images. Among them, Mip-Splatting with GS-IR (MS-IR) achieves the highest visual fidelity, accurately rendering fine anatomical structures such as pelvis contours and spine boundaries with detail comparable to the ground truth.

**Quantitative Results.** Table 1 presents the quantitative comparison on KiTS23. The standard 3DGS achieves

Table 1. Quantitative results on KiTS23 (CT) and IXI (MRI). Each group compares a baseline with its GS-IR variant. The higher performance for each metric is shown in bold.

| Dataset | Metric | 3DGS | | 2DGS | | Mip-Splatting | |
|---|---|---|---|---|---|---|---|
| | | Baseline | 3DGS-IR (Ours) | Baseline | 2DGS-IR (Ours) | Baseline | MS-IR (Ours) |
| CT | PSNR↑ | 15.20 | **16.53** | 5.472 | **19.33** | 26.23 | **28.12** |
| | SSIM↑ | **0.6242** | 0.6176 | 0.1481 | **0.6889** | 0.8603 | **0.9125** |
| | LPIPS↓ | 0.6553 | **0.3421** | 0.6761 | **0.2768** | 0.1963 | **0.0625** |
| | MSE↓ | 0.0412 | **0.0223** | 0.2845 | **0.0118** | 0.0025 | **0.0016** |
| MRI | PSNR↑ | 12.59 | **34.11** | 9.282 | **20.30** | 24.28 | **24.33** |
| | SSIM↑ | 0.4887 | **0.9695** | 0.4061 | **0.7516** | 0.8561 | **0.8575** |
| | LPIPS↓ | 0.6488 | **0.0242** | 0.4361 | **0.1852** | 0.0922 | **0.0912** |
| | MSE↓ | 0.0567 | $\mathbf{3.971 \times 10^{-4}}$ | 0.1183 | **0.0094** | 0.0039 | **0.0039** |

only 15.20 dB PSNR, and 2DGS demonstrates significantly lower performance with 5.472 dB PSNR, confirming that conventional Gaussian splatting methods cannot reconstruct internal structure from volumetric data without the proposed plane-based selection.

Applying GS-IR yields consistent improvements across all three baselines. The largest gain is observed for 2DGS: GS-IR increases PSNR by +13.86 dB (from 5.472 to 19.33 dB) and SSIM by +0.5408 (from 0.1481 to 0.6889). The best overall performance is achieved by MS-IR, reaching 28.12 dB PSNR and 0.9125 SSIM. Compared to Mip-Splatting, MS-IR reduces LPIPS from 0.1963 to 0.0625 and MSE from 0.0025 to 0.0016.

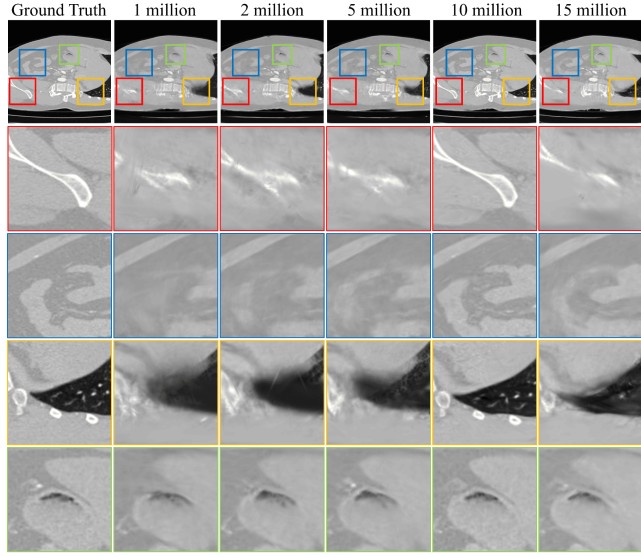

Figure 4. Qualitative comparison across different initial point cloud densities on KiTS23 (CT) using MS-IR. The top-left image is the ground truth. The remaining images correspond to 1M, 2M, 5M, 10M, and 15M initial points, respectively. The second to fifth rows show enlarged regions indicated by the colored outlines.

### 4.3. Internal View Synthesis on IXI (MRI)

**Qualitative Results.** Fig. 3 presents the visual comparison on the IXI, reinforcing the observations from the CT experiments. The standard 3DGS model produces a hazy output with severe artifacts, and the Mip-Splatting suffers from significant blurring that obscures fine anatomical detail. With GS-IR applied, all three models produce substantially improved results. The most notable improvement is achieved by 3DGS-IR which generates cross-sectional images that are visually close to the ground truth, with fine brain structures clearly resolved.

**Quantitative Results.** Table 1 reports the quantitative comparison on IXI. The standard 3DGS baseline achieves only 12.59 dB PSNR. With GS-IR, the same architecture reaches 34.11 dB, an improvement of +21.52 dB in PSNR and +0.4808 in SSIM (from 0.4887 to 0.9695). LPIPS decreases from 0.6488 to 0.0242, and MSE decreases from 0.0567 to $3.971 \times 10^{-4}$. GS-IR also improves 2DGS by +11.02 dB (from 9.282 to 20.30 dB). These results, combined with those on KiTS23, demonstrate that GS-IR enables GS methods to reconstruct internal structure across different imaging modalities.

### 4.4. Ablation Studies

#### 4.4.1. Initial Point Cloud Density

Table 2 reports the effect of varying the number of initial Gaussians from 1M to 15M. On KiTS23, the optimal performance is achieved at 10M, with 28.12 dB PSNR and 0.9125 SSIM. Reducing the density to 2M decreases PSNR by 3.09 dB and SSIM by 0.0671, as the sparser initialization leaves regions of the 3D volume under-represented. On IXI, the optimal density is 5M, achieving 24.40 dB PSNR and 0.8576 SSIM.

Fig. 4 provides a qualitative comparison across densities. Performance also degrades at 15M for both datasets. An excessively dense initialization introduces redundant Gaussians that interfere with one another during optimization, preventing convergence to a high-quality reconstruction. These results demonstrate that our initial density selection

Table 2. Effect of initial point cloud density on reconstruction quality using MS-IR. Columns indicate the number of initial Gaussians. MSE values are scaled by $10^{-3}$. Bold indicates the best result per metric.

| Dataset | Metric | 1 million | 2 million | 5 million | 10 million | 15 million |
|---|---|---|---|---|---|---|
| CT | PSNR↑ | 25.80 | 25.03 | 25.975 | **28.12** | 26.10 |
| | SSIM↑ | 0.8439 | 0.8454 | 0.8468 | **0.9125** | 0.8489 |
| | LPIPS↓ | 0.2300 | 0.2279 | 0.2223 | **0.0625** | 0.2174 |
| | MSE($\times 10^{-3}$)↓ | 2.754 | 2.714 | 2.647 | **1.647** | 2.543 |
| MRI | PSNR↑ | 23.72 | 24.03 | **24.40** | 24.33 | 23.21 |
| | SSIM↑ | 0.8460 | 0.8528 | **0.8576** | 0.8575 | 0.8162 |
| | LPIPS↓ | 0.0994 | 0.0947 | **0.0909** | 0.0912 | 0.1195 |
| | MSE($\times 10^{-3}$)↓ | 5.319 | 4.276 | **3.857** | 3.998 | 13.14 |

strategically maximizes spatial coverage while preserving computational efficiency.

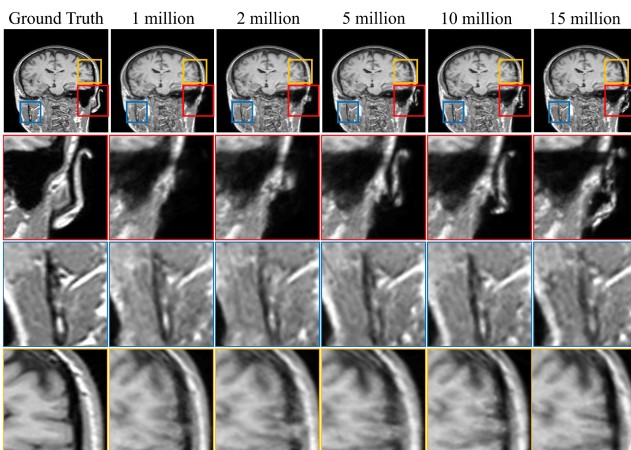

Figure 5. Qualitative comparison across different initial point cloud densities on IXI (MRI) using MS-IR. The top-left image is the ground truth. The remaining images correspond to 1M, 2M, 5M, 10M, and 15M initial Gaussians, respectively. The second to fourth rows show enlarged regions indicated by the colored outlines.

### 4.4.2. Plane Thickness

Table 3 analyzes the impact of the thickness of the plane $T$. On KiTS23, $T = 0.09$ yields the best overall performance and matches the value predicted by Eq. 1. Reducing $T$ to 0.045 reduces PSNR by 2.15 dB, and SSIM by 0.0659, indicating that an overly thin plane selects too few Gaussians per viewpoint and produces sparse, incomplete cross-sectional slices. Increasing $T$ to 0.135 also degrades performance, suggesting that a thicker plane introduces overlap with adjacent planes that cause spatial inconsistency in the gradient flow. In IXI, the metrics vary marginally across the three values, indicating that the reconstruction quality is relatively robust to $T$ within this range. Based on these results, we use $T = 0.09$ as the default setting in the remaining experiments.

Table 3. Effect of plane thickness $T$ on reconstruction quality. Bold indicates the best result per metric.

| Dataset | Metric | 0.045 | 0.09 | 0.135 |
|---|---|---|---|---|
| CT | PSNR↑ | 25.97 | **28.12** | 26.25 |
| | SSIM↑ | 0.8466 | **0.9125** | 0.8603 |
| | LPIPS↓ | 0.2245 | **0.0625** | 0.1960 |
| | MSE($\times 10^{-3}$)↓ | 2.650 | **1.620** | 2.581 |
| MRI | PSNR↑ | 24.32 | **24.33** | 24.32 |
| | SSIM↑ | **0.8584** | 0.8575 | 0.8577 |
| | LPIPS↓ | 0.0919 | 0.0912 | **0.0909** |
| | MSE($\times 10^{-3}$)↓ | 3.903 | 3.911 | **3.728** |

### 4.4.3. Number of Virtual Cameras

We analyze the dependence of reconstruction quality on the number of virtual cameras $N$ and the results are shown in Table 4. On KiTS23, the 100-camera configuration achieves the best performance with 28.12 dB PSNR and 0.9125 SSIM. Increasing $N$ to 400 degrades performance, reducing PSNR by 5.56 dB (from 28.12 to 22.56 dB) and SSIM by 0.1080 (from 0.9125 to 0.8045). On IXI, the 200-camera configuration achieves the highest PSNR of 24.94 dB, whereas the 400-camera configuration degrades performance with PSNR lowering to 23.01 dB.

This trend arises from the coupling between $N$ and the thickness of the plane $T$ defined in Eq. 1. As $N$ increases, $T$ decreases proportionally to prevent overlap between planes. A narrow plane selects fewer Gaussians per viewpoint, and therefore each viewpoint receives insufficient Gaussian coverage to render a high-quality cross-sectional image. The optimal $N$ reflects the balance between angular resolution and per-plane Gaussian coverage.

### 4.5. Novel View Synthesis via Plane Rotation

To evaluate whether GS-IR learns a continuous 3D representation rather than memorizing the discrete training views, we construct novel test viewpoints by rotating the plane between two adjacent training orientations. As illustrated in Fig. 6, given two adjacent training planes $n$ and $n+1$ separated by an angular interval $\phi$, we generate inter-

Table 4. Effect of the number of virtual cameras on reconstruction quality. Bold indicates the best result per metric.

| Dataset | Metric | 100 | 200 | 400 |
|---------|--------|-----|-----|-----|
| CT | PSNR↑ | **28.12** | 26.41 | 22.56 |
| | SSIM↑ | **0.9125** | 0.8379 | 0.8045 |
| | LPIPS↓ | **0.0625** | 0.1186 | 0.1332 |
| | MSE($\times 10^{-3}$)↓ | **1.620** | 2.400 | 7.142 |
| MRI | PSNR↑ | 24.33 | **24.94** | 23.01 |
| | SSIM↑ | 0.8575 | **0.8594** | 0.8200 |
| | LPIPS↓ | 0.0912 | **0.0897** | 0.1092 |
| | MSE($\times 10^{-3}$)↓ | 3.911 | **3.652** | 6.008 |

mediate views by rotating the $n$-th plane toward the $(n+1)$-th plane in increments of $\theta = 0.1\phi$, yielding test views at 10%, 20%, 30%, 40%, and 50% of the angle between adjacent planes. The corresponding ground-truth images are generated by trilinear re-slicing at the rotated orientation. We compare MS-IR with Mip-Splatting.

On IXI, GS-IR outperforms the baseline at rotation offsets from 20% to 50%. The advantage is most evident at 50% rotation, where GS-IR achieves 20.47 dB PSNR compared to 19.48 dB for the baseline (+0.99 dB), indicating better interpolation at the midpoint between training views. While reconstruction quality degrades for both models as the rotation offset increases, GS-IR exhibits a slower rate of degradation, maintaining an average SSIM of 0.7770 across all offsets compared to 0.7676 for the baseline. This indicates that the plane-based extraction mechanism encourages Gaussians to learn spatially consistent representations across adjacent viewpoints. The learned Gaussians generalize to unseen orientations instead of overfitting to the discrete training views.

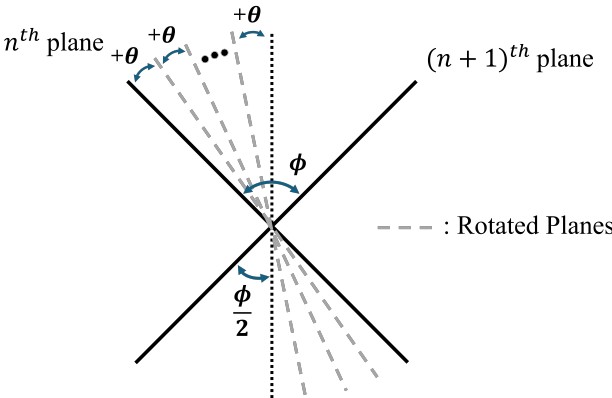

Figure 6. Illustration of the plane rotation evaluation. The view is along the $z$-axis. Solid lines represent two adjacent training planes separated by angle $\phi$. Dashed lines indicate novel test views generated by rotating the plane by increments of $\theta = 0.1\phi$.

## 5. Discussion

Medical rendering methods such as MedNeRF, X-Gaussian, and R²-Gaussian are not directly comparable to our setting. These methods target projection-based image formation, where each pixel corresponds to a line integral along a viewing ray, whereas our task requires synthesizing an anatomically localized slice on a prescribed plane. Because these settings rely on different forward models, direct quantitative comparison would require modifying their rendering equations rather than simply retraining them under our method. We therefore restrict our quantitative comparisons to Gaussian-splatting backbones under the same slice-supervision setting. More generally, GS-IR is not tied to a specific anatomy or modality: it learns internal Gaussians from volumetric data and can be trained on other datasets that contain internal structure.

Several limitations remain and suggest clear directions for future work. All volumetric planes currently pass through the volume center, which requires a centering pre-processing step for anatomy not aligned with the volume center. In addition, training with more diverse camera layouts could enable novel view synthesis from arbitrary viewpoints. The replacement of alpha-blending with a modality-aware image-formation model could further improve the reconstruction accuracy.

## 6. Conclusion

We introduce GS-IR, a plane-conditioned framework that extends 3DGS from surface-oriented rendering to internal cross-sectional representation. In standard 3DGS, optimization driven by images from visible viewpoints causes gradients to concentrate on exterior surfaces while providing no supervision to interior Gaussians. GS-IR addresses this limitation by associating each virtual camera with a volumetric plane and rendering the Gaussians selected within the plane. The selected Gaussian subset is optimized, enabling direct supervision of the internal structure. GS-IR remains simple and can be integrated into existing 3DGS variants. Our model suggests that revisiting the rendering strategy is a promising framework for extending 3DGS-based rendering beyond surface-dominant scene modeling.

We evaluate GS-IR on KiTS23 and IXI where our proposed model improves most metrics across most baselines. Our method achieves gains of up to +21.52 dB PSNR for 3DGS on IXI and +13.86 dB for 2DGS on KiTS23. In addition, novel-view evaluation via plane rotation shows that GS-IR learns spatially coherent representations that generalize to unseen orientations between training viewpoints. These results indicate that plane-conditioned supervision is an effective approach to adapt 3DGS to internal cross-sectional reconstruction in various imaging modalities.

# Acknowledgments

This work was supported by the National Research Foundation of Korea (NRF) grant funded by the Korean government (MSIT) (RS-2024-00337250).

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
