# Volumetric Plane-based Rendering: A Novel Approach for Internal View Synthesis in 3D Gaussian Splatting

## Supplementary Material

## A. Point Cloud Initialization

We observe that initializing 10 million Gaussians within the normalized coordinate range $[-1, 1]$ makes training unstable because it causes overly aggressive densification. With the same number of Gaussians distributed within a compact spatial range, the initial representation becomes excessively dense, and even modest updates to Gaussian positions produce large image-space errors. As a result, the position gradients frequently exceed the fixed densification threshold used in 3DGS, repeatedly triggering densification and exhausts GPU memory. Expanding the initialization range to $[-3, 3]$ spreads the Gaussians over a larger volume, reduces the effective initial density and stabilizes the gradient scale seen by the adaptive density control. Consequently, 10 million Gaussians are initialized randomly within this range.

## B. Plane Rotation Evaluation

Table A summarizes novel-view synthesis under plane rotation, where the baseline denotes Mip-Splatting trained without GS-IR. To evaluate interpolation between adjacent training orientations, we generate novel test planes by rotating a training slicing plane to unseen intermediate orientations, with offsets ranging from $\theta$ to $5\theta$. On KiTS23 (CT), GS-IR consistently outperforms the baseline over rotation offsets from 10% to 40%, achieving higher PSNR and SSIM together with lower LPIPS and MSE. At a 10% rotation offset, GS-IR improves PSNR from 24.97 dB to 27.74 dB (+2.77 dB) and reduces LPIPS from 0.2045 to 0.1167. On IXI (MRI), GS-IR achieves higher PSNR than the baseline from 20% to 50%, with the largest gain at 50% (+0.99 dB), indicating stronger interpolation between adjacent training orientations. These results show that GS-IR yields more robust novel-view synthesis via plane rotation across both CT and MRI.

Table A. Novel-view synthesis via plane rotation. Each column corresponds to a rotation offset from $\theta$ to $5\theta$, i.e., 10% to 50% of the angular spacing between adjacent training planes defined in Fig. 6. The baseline denotes Mip-Splatting.

| Model | Dataset | Metric | 10% | 20% | 30% | 40% | 50% |
|---|---|---|---|---|---|---|---|
| **MS-IR (Ours)** | CT | PSNR↑ | **27.74** | **25.30** | **24.03** | **23.29** | 21.59 |
| | | SSIM↑ | **0.8776** | **0.8348** | **0.8094** | **0.7931** | 0.7737 |
| | | LPIPS↓ | **0.1167** | **0.1490** | **0.1705** | **0.1841** | 0.2535 |
| | | MSE($\times10^{-3}$)↓ | **1.865** | **3.275** | **4.391** | **5.193** | 8.502 |
| | MRI | PSNR↑ | 23.15 | **21.80** | **20.77** | **20.07** | **20.47** |
| | | SSIM↑ | **0.8286** | **0.7910** | 0.7647 | 0.7460 | **0.7549** |
| | | LPIPS↓ | 0.1005 | 0.1134 | 0.1238 | 0.1320 | **0.1279** |
| | | MSE($\times10^{-3}$)↓ | 6.018 | **6.938** | **7.835** | **8.554** | **7.286** |
| **Baseline** | CT | PSNR↑ | 24.97 | 23.77 | 22.88 | 22.18 | **21.64** |
| | | SSIM↑ | 0.8408 | 0.8173 | 0.7993 | 0.7853 | **0.7744** |
| | | LPIPS↓ | 0.2045 | 0.2180 | 0.2310 | 0.2423 | **0.2514** |
| | | MSE($\times10^{-3}$)↓ | 3.351 | 4.627 | 5.927 | 7.201 | **8.432** |
| | MRI | PSNR↑ | **23.25** | 21.08 | 20.55 | 19.91 | 19.48 |
| | | SSIM↑ | 0.8174 | 0.7846 | **0.7680** | **0.7532** | 0.7149 |
| | | LPIPS↓ | **0.0993** | **0.1109** | **0.1165** | **0.1232** | **0.1279** |
| | | MSE($\times10^{-3}$)↓ | **5.881** | 7.196 | 8.241 | 8.876 | 9.150 |