# OpenReview forum: "Volumetric Plane-based Rendering: A Novel Approach for Internal View Synthesis in 3D Gaussian Splatting"
_thecvf.com/CVPR/2026/Workshop/3D4S — CVPR 2026 Workshop 3D4S Poster_

### Official Review · Reviewer_BkCN · 2026-04-13
**GS capture specifically designed for CT and MRI where partial internal data is available**

**Rating:** 6
**Confidence:** 4

**Review:**

The authors propose training with virtual cameras and multiple internal planes with selective masking and per plane adaptive density control. Their method can be applied to existing variants of GS capture like 3DGS, 2DGS, MS etc. They achieve compelling results by adding this additional plane conditioned supervision and also run ablations on different datasets, plane thicknesses, number of sampled points etc. Overall, the method is a simple extension to existing methods to leverage additional internal data to improve the capture. This approach is not scalable to other regular datasets without similar internal evidence. Additionally as the authors also report that the hyperparameters can be dataset specific, they could also extend their method to develop an automated way to select hyperparameters either embedded in the training or as an optimized search

---

### Official Review · Reviewer_i8SX · 2026-04-17
**Volumetric Plane-based Rendering: A Novel Approach for Internal View Synthesis in 3D Gaussian Splatting**

**Rating:** 7
**Confidence:** 4

**Review:**

This paper proposes a plane-based factorization of volumetric scene representations for neural rendering. Instead of relying on dense voxel grids or implicit multilayer perceptrons (MLPs) as in Neural Radiance Fields (NeRF), the method decomposes a 3D field into a set of learnable 2D planes. Rendering is performed via differentiable sampling and aggregation across these planes, aiming to achieve improved computational efficiency and memory usage while maintaining competitive visual fidelity.

Formally, the volumetric field $F(\mathbf{x})$ is approximated as:
$$
F(\mathbf{x}) \approx \sum_{i=1}^{K} f_i\big(\Pi_i(\mathbf{x})\big),
$$
where $\Pi_i(\mathbf{x})$ denotes the projection of a 3D point $\mathbf{x} \in \mathbb{R}^3$ onto the $i$-th plane, and $f_i$ represents a learnable function defined over that plane.

The overall technical quality of the paper is solid. The proposed method is well-motivated and grounded in existing literature on neural rendering and factorized representations. The formulation is mathematically consistent, and the rendering pipeline appears differentiable and implementable within standard deep learning frameworks. The experimental section demonstrates competitive performance compared to baseline methods, particularly in terms of efficiency. However, the empirical evaluation could be strengthened with more comprehensive ablation studies and robustness analyses. Strong technical quality with room for deeper experimental validation. The paper is generally understandable for readers familiar with neural rendering and 3D vision.
- The motivation for choosing plane-based decomposition is not sufficiently emphasized early on.
- Key design decisions (e.g., number of planes, resolution, aggregation strategy) are not always justified.
- Some sections are dense and assume strong prior knowledge.

Improving the exposition, particularly the contributions and intuition, would significantly enhance readability. Moderately clear, but could benefit from refinement. The originality of the work is moderate. The idea of decomposing volumetric representations into lower-dimensional structures is not entirely new. Prior works have explored the following:
- Tensor factorization of volumetric grids
- Tri-plane representations
- Hybrid explicit-implicit neural fields

The contribution lies in the specific formulation and implementation of plane-based aggregation, along with efficiency gains. This is meaningful but somewhat incremental. The significance depends largely on efficiency improvements. If the method consistently reduces computational complexity and memory usage while maintaining comparable rendering quality, it has practical value for the following:
- Real-time rendering systems
- Edge or resource-constrained environments
- Large-scale 3D reconstruction pipelines

However, the broader impact is limited by incremental novelty and lack of deeper theoretical insights. Practically significant, though not transformative.
- Efficient representation reducing memory and computation
- Scalable design compatible with modern hardware
- Fully differentiable rendering pipeline
- Competitive empirical performance
- Intuitive geometric interpretation

Weaknesses:
- Limited novelty relative to prior work
- Insufficient theoretical analysis
- Incomplete ablation studies (e.g., number of planes $K$)
- Some clarity and exposition issues
- Limited evaluation scope

This paper presents a well-executed and practically relevant approach to efficient neural rendering via plane-based volumetric decomposition. The method is technically sound and offers clear efficiency benefits, but originality is moderate and presentation could be improved.

---

### Official Review · Reviewer_btSE · 2026-04-24
**The paper introduces GS-IR, a novel training framework that adapts 3D Gaussian Splatting (3DGS) for internal volumetric reconstruction. By restricting the standard 3DGS alpha-blending pipeline to subsets of Gaussians intersecting specific slicing planes, the method successfully learns dense internal anatomical structures (e.g., CT/MRI) rather than defaulting to surface-level geometry.**

**Rating:** 6
**Confidence:** 4

**Review:**

This paper addresses the limitation of standard 3D Gaussian Splatting (3DGS) in capturing internal volumetric structures, as its gradient updates are heavily surface-biased. To resolve this, the authors propose Gaussian Splatting for Internal Representation (GS-IR). Instead of exterior view supervision, GS-IR initializes virtual cameras along a 180-degree semicircular trajectory, where each camera defines a thick volumetric plane passing through the origin. A binary mask isolates only the Gaussians falling within this plane's thickness. This selected subset is rasterized using standard alpha-blending and supervised against ground-truth cross-sectional slices extracted from the original CT/MRI volume via trilinear interpolation. By cycling through these planes, the method optimizes internal Gaussians effectively. The approach is evaluated on the KiTS23 (CT) and IXI (MRI) datasets, demonstrating massive improvements over unmodified 3DGS, 2DGS, and Mip-Splatting baselines.EvaluationQuality: The empirical validation is strong. The reported improvements are massive (e.g., up to +21.52 dB PSNR on IXI over the 3DGS baseline). This dramatic leap is expected, as standard 3DGS is fundamentally ill-equipped for slice-based internal rendering, but it effectively proves the utility of the GS-IR masking technique. The ablation studies on point cloud density, plane thickness ($T$), and camera count ($N$) are comprehensive and well-analyzed.
Clarity: The paper is highly readable and logically structured. The methodology is straightforward, and the geometric definitions for plane thickness $T = (L/2) \cdot \sin(180^\circ/N)$ and the masking operation $M_g$ are intuitive and clearly defined.
Originality: Applying a plane-based binary mask to selectively route gradients to internal Gaussians is a simple, elegant, and original solution. It neatly bypasses the need to write custom volumetric integration rasterizers by cleverly repurposing the existing tile-based 2D rasterizer for thick slice rendering.
Significance: The work demonstrates that explicit Gaussian representations can encode dense internal volumes, not just surface light fields. However, its practical clinical significance is somewhat bounded by the problem formulation: the model trains on slices generated from an already fully reconstructed 3D volume. Therefore, it functions primarily as an implicit representation/compression technique rather than a solution for sparse-view tomographic reconstruction (like R2-Gaussian).
Pros: Plug-and-Play Simplicity: The masking mechanism is agnostic to the underlying splatting architecture, allowing it to seamlessly improve 3DGS, 2DGS, and Mip-Splatting without modifying their core CUDA rasterizers.Dramatic Performance Gains: Successfully forces Gaussians to encode internal anatomy, turning failed baseline reconstructions into high-fidelity volumetric representations.Interpolation Capabilities: The novel view synthesis experiment (plane rotation) proves the model learns a spatially coherent 3D representation rather than merely overfitting to the discrete 2D training slices.Principled Hyperparameters: The derivation of plane thickness $T$ as a function of the angular separation and volume radius is a highly effective, analytically sound design choice.
Cons: Constrained Trajectory: Forcing all volumetric planes to pass through the center of the volume severely limits the spatial diversity of the slicing. Anatomical features situated far from the origin may suffer from projection artifacts or lower angular resolution compared to central features.Motivation / Use Case: Because the method requires a dense, fully realized 3D volume to generate the ground-truth trilinear slices for supervision, the practical use case is slightly circular. It would be significantly more impactful if adapted to train directly from 2D sparse-view projection data or unaligned clinical ultrasound sweeps.Sensitivity to Initialization: The ablation studies reveal that the method is highly sensitive to the initial point cloud density (e.g., dropping from 10M to 2M Gaussians in CT drops PSNR by ~3 dB). This indicates that the adaptive density control (densification/pruning) within the masked planes might not be fully sufficient to recover from poor initializations.

---

### Decision · Program_Chairs · 2026-04-28

Accept (Poster)